# Cbl Negatively Regulates NLRP3 Inflammasome Activation through GLUT1-Dependent Glycolysis Inhibition

**DOI:** 10.3390/ijms21145104

**Published:** 2020-07-19

**Authors:** Hsin-Chung Lin, Yu-Jen Chen, Yau-Huei Wei, Yu-Ting Chuang, Su-Heng Hsieh, Jing-Yu Hsieh, Yi-Lin Hsieh, David M. Ojcius, Kuo-Yang Huang, I.-Che Chung, Sheng-Ning Yuan, Yu-Sun Chang, Lih-Chyang Chen

**Affiliations:** 1Graduate Institute of Medical Sciences, National Defense Medical Center, Taipei 114, Taiwan; hsinchunglin@gmail.com; 2Division of Clinical Pathology, Department of Pathology, Tri-Service General Hospital, Taipei 114, Taiwan; 3Department of Radiation Oncology, MacKay Memorial Hospital, New Taipei City 251, Taiwan; chenmdphd@gmail.com; 4Department of Medical Research, MacKay Memorial Hospital, New Taipei City 251, Taiwan; 5Department of Nursing, MacKay Junior College of Medicine, Nursing, and Management, Taipei 112, Taiwan; 6Center for Mitochondrial Medicine and Free Radical Research, Changhua Christian Hospital, Changhua 50046, Taiwan; yhweibabi@gmail.com; 7Department of Medicine, Mackay Medical College, New Taipei City 252, Taiwan; connie801210@gmail.com (Y.-T.C.); cat12013@gmail.com (S.-H.H.); ging131307@gmail.com (J.-Y.H.); itserichsieh@gmail.com (Y.-L.H.); 8Department of Biomedical Sciences, University of the Pacific, Arthur Dugoni School of Dentistry, San Francisco, CA 94103, USA; dojcius@pacific.edu; 9Graduate Institute of Pathology and Parasitology, National Defense Medical Center, Taipei 114, Taiwan; cguhgy6934@gmail.com; 10Molecular Medicine Research Center, Chang Gung University, Taoyuan 333, Taiwan; ycc0311@gmail.com (I.-C.C.); ishucab@gmail.com (S.-N.Y.); ysc@mail.cgu.edu.tw (Y.-S.C.)

**Keywords:** Cbl, NLRP3, inflammasome, glycolysis, GLUT1

## Abstract

Activation of the nod-like receptor 3 (NLRP3) inflammasomes is crucial for immune defense, but improper and excessive activation causes inflammatory diseases. We previously reported that Cbl plays a pivotal role in suppressing NLRP3 inflammasome activation by inhibiting Pyk2-mediated apoptosis-associated speck-like protein containing a CARD (ASC) oligomerization. Here, we showed that Cbl dampened NLRP3 inflammasome activation by inhibiting glycolysis, as demonstrated with Cbl knockout cells and treatment with the Cbl inhibitor hydrocotarnine. We revealed that the inhibition of Cbl promoted caspase-1 cleavage and interleukin (IL)-1β secretion through a glycolysis-dependent mechanism. Inhibiting Cbl increased cellular glucose uptake, glycolytic capacity, and mitochondrial oxidative phosphorylation capacity. Upon NLRP3 inflammasome activation, inhibiting Cbl increased glycolysis-dependent activation of mitochondrial respiration and increased the production of reactive oxygen species, which contributes to NLRP3 inflammasome activation and IL-1β secretion. Mechanistically, inhibiting Cbl increased surface expression of glucose transporter 1 (GLUT1) protein through post-transcriptional regulation, which increased cellular glucose uptake and consequently raised glycolytic capacity, and in turn enhanced NLRP3 inflammasome activation. Together, our findings provide new insights into the role of Cbl in NLRP3 inflammasome regulation through GLUT1 downregulation. We also show that a novel Cbl inhibitor, hydrocortanine, increased NLRP3 inflammasome activity via its effect on glycolysis.

## 1. Introduction

Nod-like receptor 3 (NLRP3) inflammasome is a cytoplasmic multiprotein complex that is crucial for innate immunity. It is composed of a cytosolic pattern recognition receptor, NLRP3, along with the adaptor protein, apoptosis-associated speck-like protein containing a CARD (ASC), and pro-caspase-1 [1]. Nod-like receptor 3 inflammasome assembly is initiated by NLRP3 oligomerization, which recruits ASC to activate caspase-1, and in turn mediates the secretion of proinflammatory cytokines interleukin (IL)-1β and IL-18 [2]. Nod-like receptor 3 is a critical component required for inflammasome activation in response to some pathogen-associated molecular patterns (PAMPs) and damage-associated molecular patterns (DAMPs), such as nigericin, a pore-forming toxin derived from *Streptomyces hygroscopicus* [3]. Mice deficient in NLRP3 are very susceptible to microbial infection [4]. In addition, increasing evidence suggests that improper and excessive activation is responsible for the pathogenesis of several inflammation-associated diseases, including type 2 diabetes [5], septic shock [6], gout [7], atherosclerosis [8], rheumatoid arthritis [9], Alzheimer’s disease [10], cryopyrin-associated periodic syndrome [11], and cancer [12].

Several lines of evidence support the notion that reactive oxygen species (ROS) contribute to NLRP3 inflammasome activation [13,14,15]. Mitochondria are the major source of cellular ROS, which are generated as by-products of oxidative metabolism. Depletion of mitochondrial DNA (mtDNA) with chronic ethidium bromide treatment reduces mitochondrial reactive oxygen species (mtROS) production and inhibits NLRP3 inflammasome activation in the J774A.1 macrophage cell line [14]. Blocking the electron transport chain by using mitochondrial complex I inhibitor rotenone [16] or complex III inhibitor antimycin A [17] induces mtROS production. This enhancement of mtROS production is sufficient for activating NLRP3 inflammasomes, which suggests that mtROS is an activator of NLRP3 inflammasomes [15]. Nigericin induces mtROS production and NLRP3 inflammasome activation in macrophages, whereas treatment with mito-TEMPO, a mitochondria-specific ROS scavenger, can block NLRP3 inflammasome activation [18]. Furthermore, nigericin-induced mtROS production results in the induction and release of oxidatively modified mtDNA into the cytosol, where it binds to and activates NLRP3 inflammasomes [19]. Activation of the NLRP3 inflammasomes by oxidized mtDNA was supported in a recent study by Zhong et al., which showed that increasing mtDNA due to new synthesis enhanced NLRP3 inflammasome activation [20].

Emerging evidence also indicates that glycolysis is essential for NLRP3 inflammasome activation in macrophages [21,22]. Glycolysis converts glucose into pyruvate, which is used to yield energy for the cell during aerobic respiration through mitochondrial oxidative phosphorylation (OXPHOS) or during anaerobic lactic acid fermentation. Hexokinase 1, the enzyme of first step in glycolysis, catalyzes glucose to produce glucose-6-phosphate, and it was shown to be required for mtROS production and NLRP3 inflammasome activation in response to adenosine triphosphate (ATP) stimulation [21]. Pyruvate kinase, the enzyme involved in the last step in glycolysis, catalyzes phosphoenolpyruvate to generate pyruvate and it is required for ATP-induced NLRP3 inflammasome activation [22].

The proto-oncogene Cbl encodes a ubiquitin ligase belonging to the Cbl family, and mediates protein ubiquitination [23]. This protein contains an N-terminal phosphotyrosine-binding domain that allows it to interact with numerous tyrosine-phosphorylated substrates, targeting them for proteasomal or lysosomal degradation. Thus, the Cbl proteins function as a negative regulator of many signal transduction pathways. Our previous report revealed that Cbl is pivotal in suppressing NLRP3 inflammasome activation in response to stimulation by nigericin or ATP by inhibition of Pyk2-dependent ASC oligomerization [24,25]. Phosphorylated Pyk2 (p-Pyk2) can directly phosphorylate ASC at Tyr146 [24]—such phosphorylation is required for ASC oligomerization and the sequential formation of the NLRP3 inflammasomes [25,26]. Through the ubiquitination-mediated proteasomal degradation of p-Pyk2, Cbl reduces the level of p-Pyk2 and inhibits NLRP3 inflammasome activation. In addition, Cbl plays a role in energy homeostasis in a Cbl-knockout (KO) mice [27]. Cbl-KO mice exhibit a profound increase in whole-body energy expenditure, as determined by increased core temperature and whole-body oxygen consumption. The Cbl-KO mice also display marked improvement in whole-body insulin action and glucose tolerance. Consistent with our previous studies, the size of the mitochondria was found to be enlarged in Cbl-deficient cells [24,27]. All these observations suggest that Cbl is involved in the regulation of energy homeostasis, although the underlying mechanism has not been identified yet.

In this study, we discovered that Cbl dampens NLRP3 inflammasome activation through glycolysis inhibition. Cbl acts through a post-transcriptional mechanism to reduce the amount of glucose transporter 1 (GLUT1) protein available for cellular glucose uptake, which in turn affects the capacities of glycolysis and mitochondrial oxidative phosphorylation (OXPHOS). The low level of glycolysis, regulated by Cbl, further dampens NLRP3 inflammasome activation. Together, our findings provide new insights into the role of the Cbl in suppressing NLRP3 inflammasome activation through glycolysis inhibition.

## 2. Results

### 2.1. Cbl Inhibition Increased the Rate of Glycolysis and Oxidative Phosphorylation

We had previously observed that Cbl acts as a negative regulator of NLRP3 inflammasomes [24] and it may be involved in the regulation of glucose tolerance [27]. Recent reports have indicated that glycolysis contributes to the activation of the NLRP3 inflammasomes [21,22,28]; therefore, we hypothesized that Cbl may dampen NLRP3 inflammasome activation by targeting glycolysis. We generated Cbl-KO THP-1 cells through clustered regularly interspaced short palindromic repeats (CRISPR)/Cas9 technology and characterized the expression of Cbl and NLRP3 inflammasome components (Figure 1A). Consistent with our previous study, decreasing Cbl expression did not affect the expression levels of NLRP3 inflammasomes [24]. To determine whether glycolysis is a Cbl target, we measured the extracellular acidification rate (ECAR), an indicator of glycolysis, in Cbl-KO and wild-type (WT) THP-1-derived macrophages. We found that Cbl-KO THP-1-derived macrophages had higher rates of glycolysis than WT cells, as reflected by their maximal ECAR when elicited by the inhibition of ATP synthase in OXPHOS with oligomycin, which revealed the glycolytic capacity (difference between maximal ECAR and non-glycolytic acidification) and glycolytic reserve capacity (difference between glycolytic capacity and basal glycolysis; Figure 1B). To examine whether an increase in glycolytic activity due to Cbl deficiency was a general effect, we assessed glycolysis in Cbl-KO human embryo kidney (HEK) 293T cells. Cbl-KO HEK293T cells with NLRP3 inflammasome reconstituted by transduction were shown previously to have higher levels of NLRP3 inflammasome activation than in NLRP3 inflammasome-reconstituted WT HEK293T cells [24]. Thus, the loss of Cbl increased glycolytic capacity and glycolytic reserve capacity in HEK293T cells (Figure 1C). Besides the increase in glycolytic activity due to genetic loss of Cbl, we examined whether glycolysis was affected by pharmacological inhibition with a Cbl-specific inhibitor, hydrocotarnine (CRIN-2, patent number: WO2011160016A2), which was shown previously to interfere with the suppression effect of Cbl on NLRP3 inflammasome-dependent IL-1β secretion [24]. As illustrated in Figure 1D, treatment with hydrocotarnine significantly increased the glycolytic capacity and glycolytic reserve capacity in THP-1-derived macrophages, compared with untreated control cells. Since glycolysis is a key metabolic pathway that feeds into mitochondrial OXPHOS, we hypothesized that an increase of glycolytic capacity due to Cbl inhibition may enhance the maximal rate of mitochondrial OXPHOS. As expected, Cbl-KO THP-1-derived macrophages (Figure 1E) or Cbl-KO HEK293T cells (Figure 1F) had higher OXPHOS than WT cells, as reflected by their maximal uncontrolled oxygen consumption rate (OCR), when elicited by the dissipation of the mitochondrial electrochemical proton gradient with the mitochondrial uncoupler carbonyl cyanide 4-(trifluoromethoxy)phenylhydrazone (FCCP), which revealed the maximum respiration (difference between maximal OCR and non-mitochondrial respiration) and spare respiratory capacity (SRC; difference between maximum respiratory capacity and basal respiratory capacity). Consistent with the genetic depletion of Cbl, Cbl inhibition with hydrocotarnine increased the maximum respiration and SRC of THP-1-derived macrophages (Figure 1G). Together, these data revealed that Cbl inhibition through either genetic depletion or pharmacological drug treatment was sufficient to increase the glycolytic capacity and OXPHOS capacity.

### 2.2. Glycolysis Upregulation by Cbl Inhibition Enhanced NLRP3 Inflammasome Activation

Because glycolysis is required for NLRP3 inflammasome activation [21,22,28], we examined whether increased glycolytic activity due to Cbl inhibition affected NLRP3 inflammasome activation. As presented in Figure 2A,B, activation of glycolysis was immediately induced by nigericin stimulation along with OXPHOS activation in WT and Cbl-KO THP-1-derived macrophages. A higher level of OXPHOS activation in Cbl-KO cells than in WT cells was correlated with a higher level of glycolysis activation (Figure 2A,B, right panels). To examine the role of glycolysis in OXPHOS activation, we inhibited glycolysis in macrophages by using 2-deoxyglucose (2DG), a potent glycolysis inhibitor [21]. Inhibition of glycolysis (Figure 2A, right panel) resulted in delayed OXPHOS activation 8 min after nigericin stimulation in WT and Cbk-KO cells (Figure 2B), which suggested that glycolytic activity is required for OXPHOS activation by nigericin. Interestingly, the level of OXPHOS inhibition caused by glycolysis inhibition was even higher in Cbl-KO cells than in WT cells (Figure 2B, right panel), which indicated that more glycolysis was used in Cbl-KO cells to fuel OXPHOS activation in response to NLRP3 inflammasome activation. Finally, we examined whether glycolysis was involved in Cbl-regulated NLRP3 inflammasome inhibition. We analyzed the effect of glycolysis inhibition with 2DG on the activation of caspase-1 and the maturation of IL-1β in Cbl-KO and WT THP-1-derived macrophages treated with nigericin. The levels of mature IL-1β p17 and active caspase-1 (as assessed by the level of caspase-1 p20) in Cbl-KO cells were higher than in WT cells but were reduced by pretreatment with 2DG or glucose deprivation, while NLRP3, pro-caspase-1, and pro-IL-1β levels were unchanged (Figure 2C). Consistent with these results, the levels of IL-1β secretion in Cbl-KO cells were also higher than that in WT cells but were inhibited by 2DG pretreatment (Figure 2D). These results suggested that increased glycolysis due to Cbl inhibition fuels more OXPHOS activation in response to NLRP3 inflammasome activation and enhances NLRP3 inflammasome activation.

### 2.3. Regulation of Cellular ROS Production by Cbl Enhanced NLRP3 Inflammasome Activation

Because cellular ROS are primarily generated as a by-product of OXPHOS and contribute to NLRP3 inflammasome activation [14,15,21], we evaluated whether ROS production is regulated by Cbl and contributes to NLRP3 inflammasome activation. In response to nigericin stimulation, both Cbl-KO and WT THP-1-derived macrophages exhibited increased production of cellular ROS (Figure 3A). Notably, higher levels of cellular ROS production were observed in Cbl-KO cells than in WT cells. To examine the role of cellular ROS production in NLRP3 inflammasome activation, cellular ROS were neutralized using the ROS scavenger N-acetyl-L-cysteine (NAC) [13]. Neutralization of cellular ROS with NAC (Figure 3B) completely inhibited the expression of active caspase-1 p20 and mature IL-1β p17 (Figure 3C) and the secretion of IL-1β (Figure 3D) in both Cbl- KO and WT cells, while NLRP3, pro-caspase-1, and pro-IL-1β levels were unchanged. These results suggested that Cbl inhibition increases the cellular ROS production, which contributes to NLRP3 inflammasome activation.

### 2.4. Inhibition of Cbl Increased Cellular Glucose Uptake

Because glycolysis is regulated by Cbl and is critical for NLRP3 inflammasome activation (Figure 2), we examined the mechanisms involved in Cbl-mediated glycolysis regulation. In a previous report, Cbl-deficient mice displayed a marked improvement in glucose tolerance [27], suggesting that Cbl inhibition may increase the transportation of glucose from outside to inside the cells, which is necessary for supplying the intracellular glucose needed to start glycolytic catabolism within cells. Therefore, we examined whether Cbl is involved in cellular glucose uptake by measuring the uptake of 2-(N-(7-Nitrobenz-2-oxa-1,3-diazol-4-yl)Amino)-2-Deoxyglucose (2-NBDG), a fluorescent analog of glucose in Cbl-KO and WT THP-1-derived macrophages. The 2-NBDG uptake in Cbl-KO cells was 1.5-fold higher than in WT cells (Figure 4A). Cbl inhibition with hydrocotarnine consistently increased 2-NBDG uptake by 1.3-fold compared with the untreated control (Figure 4B). These results suggested that inhibiting Cbl is sufficient to increase cellular glucose uptake in THP-1-derived macrophages.

### 2.5. Surface Expression of GLUT1 Protein Was Post-Transcriptionally Regulated by Cbl

Cellular glucose uptake is achieved using glucose transporters (GLUTs) 1–4. Among these transporters, GLUT1 is the most abundant glucose transporter in macrophages [29]. Because cellular glucose uptake increased through Cbl inhibition (Figure 4), we hypothesized that Cbl is involved in GLUT1 regulation. As expected, the surface expression of GLUT1 protein in Cbl-KO THP-1-derived macrophages as measured by fluorescence-activated cell sorting analysis was higher than in WT cells (Figure 5A). Hydrocotarnine-treated THP-1-derived macrophages consistently expressed a higher level of cell-surface GLUT1 protein than untreated control cells (Figure 5B). Since Cbl is an ubiquitin ligase that is involved in ubiquitin-mediated protein degradation [23], we examined whether GLUT1 is post-transcriptionally regulated by Cbl. As indicated in Figure 5C,D, neither the generic depletion of Cbl nor pharmacological inhibition of Cbl with hydrocotarnine affected Cbl mRNA expression when measured through quantitative reverse transcriptase polymerase chain reaction (RT-qPCR), which suggested that Cbl is not involved in the regulation of GLUT1 mRNA. Then, we examined whether the upregulation of cell-surface Cbl protein in Cbl-KO cells is attributed to the increase in the total GLUT1 protein. As expected, the level of total GLUT1 protein in Cbl-KO cells was higher than in WT cells (Figure 5E). The level of total GLUT1 protein in hydrocotarnine-treated cells was also higher than in untreated control cells (Figure 5F). Similarly, the total GLUT1 protein in Cbl-KO cells increased 2.7-fold compared with WT cells (Figure 5G) or 1.6-fold in hydrocotarnine-treated cells compared with the untreated control (Figure 5H), as confirmed by immunoblot analysis. Together, these results suggested that inhibiting Cbl increases surface expression of GLUT1 protein through post-transcriptional regulation.

### 2.6. Cbl Regulation of GLUT1 Modulated Cellular Glucose Uptake and NLRP3 Inflammasome Activation

Previous results suggest that increased GLUT1 expression and glycolysis due to Cbl inhibition is required for NLRP3 inflammasome activation, suggesting that GLUT1 may be responsible for Cbl-mediated NLRP3 inflammasome regulation. We therefore examined whether GLUT1 depletion affects cellular glucose uptake, discovering that the levels of cellular glucose uptake were lower in WT and Cbl-KO THP-1-derived macrophages treated with a GLUT1-specific small interfering RNA (siRNA) than in cells treated with control siRNA (Figure 6A,B). To evaluate whether GLUT1 is involved in NLRP3 inflammasome activation, we analyzed changes in caspase-1 activation, IL-1β maturation, and IL-1β secretion in cells treated with nigericin. The levels of active caspase-1 p20, mature IL-1β p17 (Figure 6C), and IL-1β secretion (Figure 6D) were reduced in GLUT1 knockdown cells. Together, these results indicated that GLUT1 protein, a target of Cbl, contributes to NLRP3 inflammasome activation.

## 3. Discussion

The activation of NLRP3 inflammasomes is critical for immune defense; however, improper and excessive activation can lead to inflammatory diseases. Much remains to be learned regarding the mechanisms that restrain inflammasome activation under normal conditions. On the basis of our present results, we proposed a new mechanism that restrains NLRP3 inflammasome activation by Cbl (Figure 7). Cbl downregulates cell-surface GLUT1, which sequentially leads to reduction of cellular glucose uptake, glycolytic capacity, OXPHOS capacity, and ROS production, which in turn dampen NLRP3 inflammasome activation. Importantly, the inhibition of GLUT1 by Cbl can be reversed by inhibiting Cbl with the pharmacological inhibitor hydrocotarnine, which therein enhances NLRP3 inflammasome activation. Together, these novel results revealed a new mechanistic role for the Cbl–GLUT1 axis in negative regulation of NLRP3 inflammasome.

Previously, we revealed that Cbl is crucial in suppressing NLRP3 inflammasome activation from stimulation by nigericin or ATP [24]. The phosphorylation of Cbl is regulated by Src-family kinases, and phosphorylation of Cbl at Tyr371 suppresses NLRP3 inflammasome activation [24]. Mechanistically, Cbl reduces the level of phosphorylated Pyk2 (p-Pyk2) through post-transcriptional regulation and ubiquitination-mediated proteasomal degradation. P-Pyk2 is required to directly phosphorylate the caspase-recruitment domain of ASC at Tyr146, which converts ASC into an oligomerization-competent state [25]. Upon NLRP3 stimulation, NLRP3 oligomers recruit ASC molecules and trigger ASC oligomerization, which is essential for activating NLRP3 inflammasomes [25]. The lower level of p-Pyk2 is regulated by Cbl, which thus dampens NLRP3 inflammasome activation. In addition to the Cbl-p-Pyk2 pathway, we identified a new mechanism underlying Cbl-regulated inhibition of NLRP3 inflammasomes. The low level of GLUT1 expression, which is regulated by Cbl through post-transcriptional regulation (Figure 5), inhibits glucose uptake and glycolytic metabolism and finally dampens NLRP3 inflammasome activation and IL-1β secretion (Figure 6). Consistent with our results, overexpression of GLUT1 in macrophages resulted in increased glucose uptake, increased glucose utilization, and increased IL-1β expression [29,30], which suggested that GLUT1 is involved in regulating IL-1β expression. In addition to the regulation of IL-1β expression, we noted that GLUT1 is involved in regulating IL-1β secretion by enhancing NLRP3 inflammasome activation. Together, our studies have indicated that Cbl is a master regulator of NLRP3 inflammasomes. Cbl can inhibit the NLRP3 inflammasomes through two pathways, by inhibition of p-Pyk2-regulated ASC oligomerization and inhibition of GLUT1-regulated glucose utilization.

Several lines of evidence support the notion that energy metabolic remodeling, including glycolysis [21,22] and fatty acid oxidation [31,32], plays a role in NLRP3 inflammation activation. In response to various NLRP3 stimuli, metabolic remodeling activates NLRP3 inflammasomes through ROS production. Although ROS is involved in many physiological functions, it also causes cellular oxidative damage and has been linked to multiple pathologies, including neurodegenerative diseases, diabetes, cancer, and premature aging [33]. Here, we noted that Cbl is involved in regulation of ROS production (Figure 3). Through GLUT1 inhibition, Cbl restrains the activation of glycolysis and mitochondrial OXPHOS, which are required for ROS production and NLRP3 inflammasome activation [21]. Even though Cbl dampens NLRP3 inflammasome activation, reduction of ROS production by Cbl may limit ROS damage to the cells and thus help to maintain the health of cells. Therefore, our results suggested a new role for Cbl in energy remodeling and energy homeostasis through the regulation of glucose utilization. In agreement with our suggestion, Cbl-KO mice exhibited higher energy expenditure and were protected from obesity, insulin resistance, and diabetes [27]. Our new findings revealed a new role for Cbl in regulation of glucose utilization.

Diabetes is a metabolic disease in which glucose levels are too high in the blood vessels but are not internalized and utilized by the cell. Our results showed that cellular glucose uptake was restrained by Cbl (Figure 4) through GLUT1 inhibition (Figure 6B), which indicated that Cbl plays a role in the regulation of glucose utilization, which may help to explain the diabetes resistance of Cbl-KO mice [27]. Diabetes is also considered as an immune dysfunction disease, and patients with diabetes have infections more often than people without diabetes [34,35]. Diabetes approximately doubles the risk of cellulitis, compared with people without diabetes [34,36]. Common organisms that cause cellulitis are *Streptococcus pyogenes* and *Staphylococcus aureus* [37,38]. The NLRP3 inflammasomes are required for protecting mice from *S. pyogenes* or *S. aureus* infections [39,40]. Our results indicated that Cbl inhibition with hydrocotarnine not only increases expression of GLUT1 (Figure 5) but also increases cellular glucose uptake (Figure 4), which enhances NLRP3 inflammasome activation (Figure 2 and Figure 6). Therefore, we purpose that inhibition of Cbl with hydrocotarnine may protect people from diabetes and contribute to the elimination of bacterial infections in diabetic people. Importantly, inhibition of Cbl with hydrocotarnine increases inflammasome-mediated IL-18 secretion in the colon and protects mice from dextran sulphate sodium-induced colitis [24]. Our present study provided new evidence to support a novel clinical use for hydrocotarnine in disease treatment.

## 4. Materials and Methods

### 4.1. Cell Culture

The THP-1 (human leukemia monocytic) cell line was purchased from the Biosource Collection and Research Center (Hsinchu, Taiwan) and maintained in Roswell Park Memorial Institute (RPMI) medium, as described previously [41]. To generate Cbl-KO THP-1 cells, we manipulated THP-1 cells with a lentiviral-based CRISPR/Cas9 system with small guide RNA (sgRNA) to target Cbl (GTCCA CCGTC CCCGG CGGGT), followed by selection with puromycin, as described previously [24]. For Cbl inhibition, THP-1 cells were maintained in RPMI containing 50 μM hydrocotarnine (enamine). For macrophage differentiation, THP-1 cells were stimulated with 200 nM phorbol 12-myristate 13-acetate for 16 h. The HEK293T cell line was obtained from American Type Culture Collection (Rockville, MD, USA) and was grown in Dulbecco’s modified Eagle’s medium supplemented with 10% (v/v) fetal bovine serum, penicillin (1000 U/mL), and streptomycin (50 μg/mL) with 5% CO_2_ at 37 °C. Cbl-KO HEK293T cells were generated as described previously [24]. For inflammasome stimulation, the cells were treated with 10 μM nigericin for 45 min. For inhibition, 100 mM of 2DG (Sigma) and NAC (25 mM) [13] were applied for 1 h and 5 min prior to inflammasome stimulation, respectively.

### 4.2. Seahorse Analysis

The Seahorse XF24 bioanalyzer (Seahorse Bioscience, Billerica, MA, USA) was used to continuously monitor the ECAR and OCR. We plated 5 × 10^4^ cells per well (THP-1 or Cbl-KO THP-1 cells) or 4 × 10^4^ cells per well (HEK293T or Cbl-KO HEK293T) on XF24 cell culture microplates (Seahorse Bioscience). ECAR or OCR, as parameters of glycolytic flux and mitochondrial respiration, were measured on a Seahorse XF24 bioanalyzer by using the Glycolysis Stress Test kit or Mito Stress Test Kit (Seahorse Bioscience) according to the manufacturer’s instructions. To induce maximum OCR, we used 1 and 0.3 μM of FCCP in THP-1 or Cbl-KO THP-1 cells and HEK293T or Cbl-KO HEK293T cells, respectively. Glycolytic capacity was calculated as the difference between maximum ECAR measurement after oligomycin addition and the last ECAR measurement before glucose addition. Glycolytic reserve capacity was calculated as the difference between the latest ECAR measurement before and maximum ECAR measurement after oligomycin addition. Maximum respiration was calculated as the difference between maximum OCR measurement after FCCP addition and minimum OCR measurement after rotenone and antimycin A addition. SRC was calculated as the difference between basal OCR and maximal OCR after the addition of FCCP.

### 4.3. Immunoblot Analysis

The procedure of immunoblot analysis was previously described [42]. To measure caspase-1 activation and IL-1β secretion, we concentrated culture supernatants using trichloroacetic acid protein precipitation, followed by SDS-PAGE separation. The following primary antibodies were used for the immunoblot: anti-ASC (1:1000; cat# SC-22514-R; Santa Cruz Biotechnology, Dallas, TX, USA), anti-caspase-1 (1:300; cat# SC-56036; Santa Cruz Biotechnology), anti-IL-1β (1:1000; SC-32294; Santa Cruz Biotechnology), anti-glyceraldegyde 3-phosphate dehydrogenase (GAPDH; 1:2000; SC-32233; Santa Cruz Biotechnology), anti-NLRP3 (1:500; cat# AG-20B-0014; Adipogen, San Diego, CA, USA), and anti-GLUT1 (1:1000; cat# ab115730; Abcam, Cambridge, United Kingdom). Secondary antibodies used for immunoblotting were as follows: goat anti-rabbit immunoglobulin horseradish peroxidase (IgG-HRP; 1:5000; cat# SC-2004; Santa Cruz Biotechnology) and sheep antimouse IgG-HRP (1:5000; cat# NA931; Amersham, Amersham, United Kingdom). The immunoreactive protein bands were visualized using the enhanced chemiluminescence system (cat# TU-ECL03; Biotools, Taiwan) according to the manufacturer’s instructions.

### 4.4. Glucose Uptake Assay

THP-1-derived macrophages were incubated in RPMI medium with 10% fetal bovine serum (FBS) and 5.5 mM glucose in the presence of 200 μM 2-NBDG for 16 h. The cellular glucose uptake was determined by the fluorescence intensity of the internalized 2-NBDG that was measured using flow cytometry.

### 4.5. Assessment of GLUT1 Expression by Flow Cytometry

For cell-surface GLUT1 detection, cells were stained with fluorescein isothiocynate-conjugated anti-GLUT1 Ab (cat# FAB1418F; R&D systems, Minneapolis, MN, USA) at 4 °C. Mouse IgG2b was used as an isotype control. For total GLUT1 detection, cells were fixed and permeabilized at 4 °C with Intracellular Fixation and Permeabilization Buffer Set (cat# 88-8824; eBioscience, San Diego, CA, USA), followed by GLUT1 staining. GLUT1 staining was measured using the FACSCalibur flow cytometer and CellQuest pro software (BD Biosciences, San Jose, CA, USA).

### 4.6. Quantitative RT-PCR

RNA samples were isolated using the TRIzol reagent (Invitrogen, Carlsbad, CA, USA), followed by reverse transcription of RNA (1 μg) using oligo(dT)20 primers (Invitrogen) and Moloney Murine Leukemia Virus Reverse Transcriptase (M-MLV RT; Invitrogen) according to the manufacturer’s instructions. The following primers were used: GLUT1—forward 5′-CTTCT CCAAC TGGAC CTCAA ATTT-3′ and reverse 5′-CGGCC TTTAG TCTCA GGAAC TTT-3′, and GAPDH (normalization control)—forward 5′-TGGTA TCGTG GAAGG ACTCA TGAC-3′ and reverse 5′-ATGCC AGTGA GCTTC CCGTT CAGC-3′. Quantitative RT-PCR was performed using a Light-Cycler Instrument (Roche Diagnostics, Basel, Switzerland) with the FastStart DNA Master SYBR Green I reagent (Roche Diagnostics).

### 4.7. RNA Interference

The double-stranded RNA duplexes targeting GLUT1 were purchased from Dharmacon, and the negative control siRNA was synthesized by Research Biolabs Ayer Rajah Industrial Estate (Research Biolabs, Singapore). They were transfected into cells using Lipofectamine 2000, as previously described [24]. For efficient knockdown, the cells were incubated for 2 days. The target sequence of GLUT1 siRNA was 5′-AGACA UGGGU CCACC GCUA-3′.

### 4.8. IL-1β Enzyme-Linked Immunosorbent Assay

Cell culture supernatants were assayed for human IL-1β (eBioscience) as described previously [43].

### 4.9. Cellular ROS Production Assay

Cellular ROS was measured using 2′,7′-dichlorodihydrofluorescein diacetate (H_2_-DCFDA; Life Technologies, Carlsbad, CA, USA) as described previously [24]. The cells were simultaneously treated with 10 μM nigericin and 0.3 μM H_2_-DCFDA for 30 min and then analyzed using flow cytometry.

### 4.10. Statistical Analysis

Statistical analyses were performed using SPSS 13.0 (SPSS Inc., Chicago, IL, USA). Differences were considered significant at *p* < 0.05. The error bars were calculated and represented in terms of mean ± SD.

## 5. Conclusions

In summary, we proposed a model for the previously unrecognized Cbl-GLUT1 axis-dependent suppression of NLRP3 inflammasome activation. We discovered that Cbl inhibited surface expression of GLUT1 protein, which in turn increased the threshold for NLRP3 inflammasome activation by maintaining glycolysis, OXPHOS, and ROS at low levels. Our findings suggested that Cbl should be considered a new therapeutic target of hydrocotarnine when treating inflammation-related diseases related to NLRP3 inflammasome activation.

## Figures and Tables

**Figure 1 ijms-21-05104-f001:**
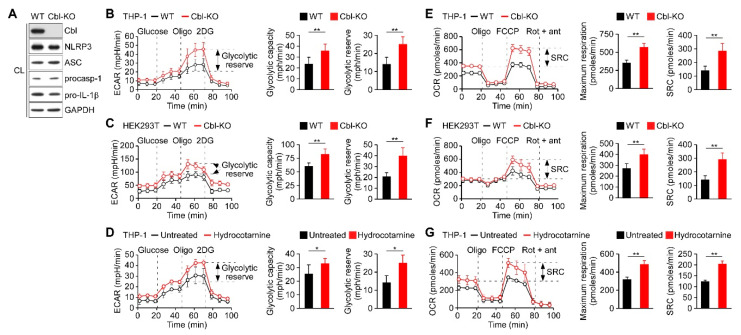
Inhibition of Cbl increased the glycolytic capacity and mitochondrial respiration. (**A**) Immunoblot of Cbl and nod-like receptor 3 (NLRP3) inflammasome molecules in wild-type (WT) and Cbl-knockout (KO) THP-1-derived macrophages; (**B**) extracellular acidification rate (ECAR) of WT and Cbl-KO THP-1-derived macrophages; (**C**) WT and Cbl-KO HEK293T cells; and (**D**) hydrocotarnine-treated and untreated THP-1-derived macrophages, under sequential treatment (dotted vertical lines) with glucose, oligomycin (Oligo), and 2-deoxyglucose (2DG). The glycolytic capacity and glycolytic reserve capacity are calculated in the right panels. (**E**) Oxygen consumption rate (OCR) of Cbl-KO THP-1-derived macrophages; (**F**) Cbl-KO HEK293T cells; and (**G**) hydrocotarnine-treated THP-1-derived macrophages, under sequential treatment (dotted vertical lines) with oligomycin, carbonyl cyanide 4-(trifluoromethoxy)phenylhydrazone (FCCP), and rotenone plus antimycin. Maximum respiration and spare respiratory capacity (SRC) are calculated in the right panels. * *p* < 0.05; ** *p* < 0.01. All results are presented as the mean ± standard deviation (SD) of the three independent experiments and were analyzed using the Student’s *t*-test.

**Figure 2 ijms-21-05104-f002:**
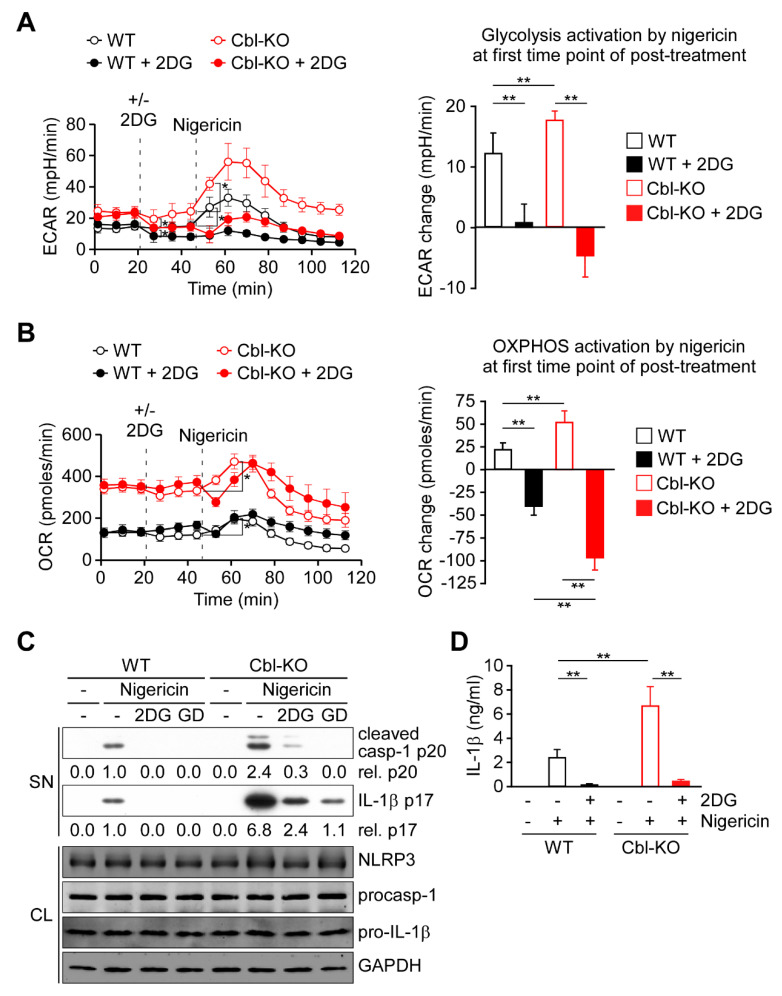
Cbl deficiency promoted NLRP3 inflammasome activation via upregulation of glycolysis. (**A**) ECAR of WT and Cbl-KO THP-1-derived macrophages under sequential treatment (dotted vertical lines) with or without 2DG and treated with nigericin. The change in ECAR between before and after nigericin treatment is calculated in the right panel. (**B**) OCR of WT and Cbl-KO THP-1-derived macrophages under sequential treatment (dotted vertical lines) with or without 2DG and treated with nigericin (**B**). The change in OCR between before and after nigericin treatment is calculated in the right panel. (**C**) Immunoblot analysis of caspase-1 (p20) and IL-1β (p17) in culture supernatants (SN), and NLRP3 inflammasome molecules and glyceraldegyde 3-phosphate dehydrogenase (GAPDH) in cell lysates (CL) of WT and Cbl-KO THP-1-derived macrophages that were treated with 2DG or glucose deprivation and then stimulated with nigericin. (**D**) Enzyme-linked immunosorbent assay of IL-1β in the supernatants of WT and Cbl-KO THP-1-derived macrophage cultures pretreated with or without 2DG and then stimulated with nigericin. ** *p* < 0.01. All results are presented as the mean ± SD of three independent experiments and were analyzed using Student’s *t*-test.

**Figure 3 ijms-21-05104-f003:**
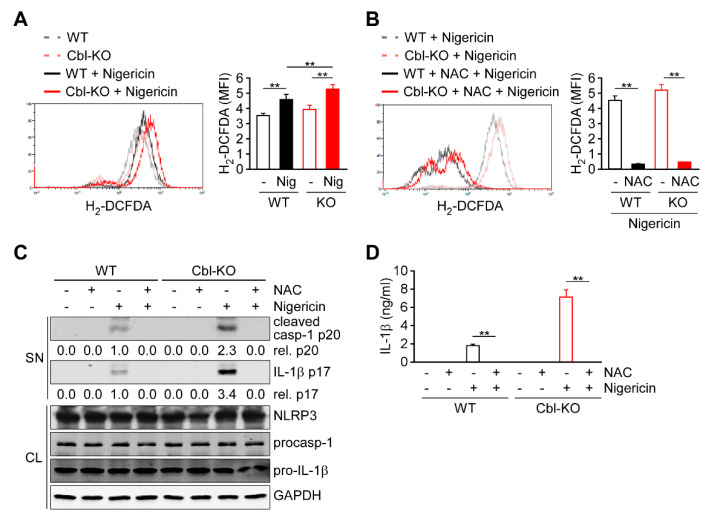
Reactive oxygen species (ROS) production increased in Cbl-deficient THP-1-derived macrophages and enhanced NLRP3 inflammasome activation. (**A**) The production of cellular ROS in WT and Cbl-KO THP-1-derived macrophages treated with or without nigericin for 30 min was measured using 2′,7′-dichlorodihydrofluorescein diacetate (H_2_-DCFDA). (**B**) The production of cellular ROS in WT and Cbl-KO THP-1-derived macrophages pretreated with or without N-acetyl-L-cysteine (NAC) and stimulated with nigericin for 30 min was measured using H_2_-DCFDA. (**C**) Immunoblot analysis of caspase-1 (p20) and interleukin (IL)-1β (p17) in culture supernatants (SN) and NLRP3 inflammasome molecules and GAPDH in cell lysates (CL) of WT and Cbl-KO THP-1-derived macrophages that were pretreated with or without NAC and then stimulated with nigericin. (**D**) Enzyme-linked immunosorbent assay of IL-1β in the supernatants of WT and Cbl-KO THP-1-derived macrophage cultures pretreated with or without NAC and then stimulated with nigericin. ** *p* < 0.01. All results are presented as the mean ± SD of three independent experiments and were analyzed using Student’s *t*-test.

**Figure 4 ijms-21-05104-f004:**
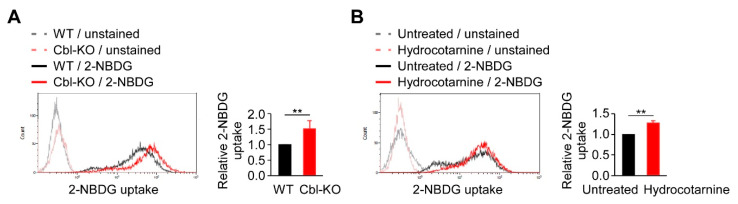
Cellular glucose uptake was found to be regulated by Cbl. (**A**) A representative flow cytometry plot of three experiments on glucose uptake in WT and Cbl-KO THP-1-derived macrophages. (**B**) A representative flow cytometry plot of three experiments on glucose uptake in hydrocotarnine-treated and untreated THP-1-derived macrophages. ** *p* < 0.01. All results are presented as the mean ± SD of three independent experiments and were analyzed using Student’s *t*-test.

**Figure 5 ijms-21-05104-f005:**
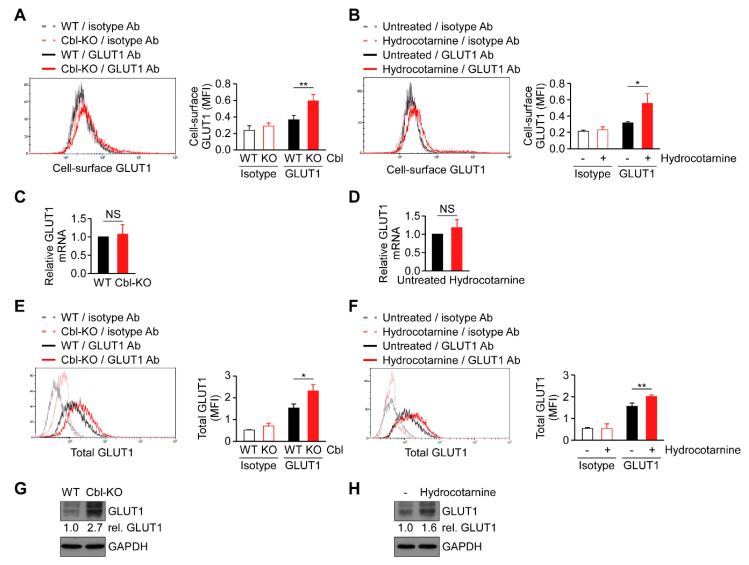
Cbl reduced glucose transporter 1 (GLUT1) protein expression through post-transcriptional regulation. Flow cytometry plots of three experiments on revealing the surface expression of GLUT1 protein in (**A**) WT and Cbl-KO THP-1-derived macrophages or in (**B**) hydrocotarnine-treated and untreated THP-1-derived macrophages. Changes in GLUT1 mRNA expression were assessed in (**C**) WT and Cbl-KO THP-1-derived macrophages or in (**D**) hydrocotarnine-treated and untreated THP-1-derived macrophages. Flow cytometry plots of three experiments on revealing the expression of total GLUT1 protein in (**E**) WT and Cbl-KO THP-1-derived macrophages or in (**F**) hydrocotarnine-treated and untreated THP-1-derived macrophages. Immunoblot analysis of GLUT1 in (**G**) WT and Cbl-KO THP-1-derived macrophages or in (**H**) hydrocotarnine-treated and untreated THP-1-derived macrophages. * *p* < 0.05; ** *p* < 0.01. All results are presented as the mean ± SD of three independent experiments and were analyzed using Student’s *t*-test.

**Figure 6 ijms-21-05104-f006:**
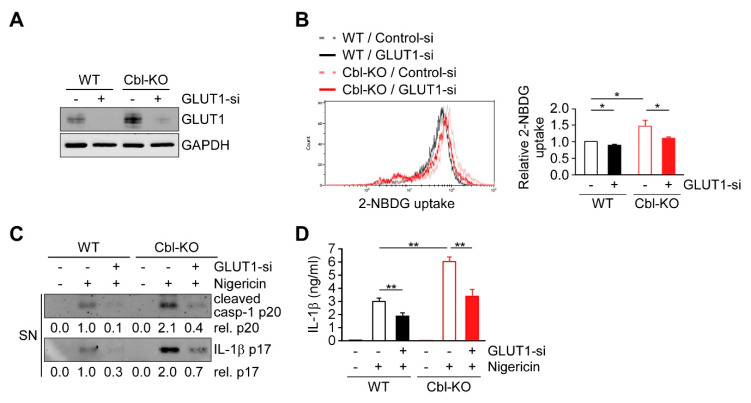
GLUT1 promoted cellular glucose uptake and NLRP3 inflammasome activation. (**A**) Immunoblot of GLUT1 in THP-1-derived macrophages treated with GLUT1 small interfering RNA (siRNA) or control siRNA. (**B**) Flow cytometry plots of three experiments showing glucose uptake in WT and Cbl-KO THP-1-derived macrophages treated with GLUT1 siRNA or control siRNA. (**C**) Immunoblot analysis of caspase-1 (p20) and IL-1β (p17) in culture supernatants of WT and Cbl-KO THP-1-derived macrophages treated with GLUT1 siRNA or control siRNA and then treated with nigericin. (**D**) Enzyme-linked immunosorbent assay of IL-1β in the supernatants of WT and Cbl-KO THP-1-derived macrophages treated with GLUT1 siRNA or control siRNA and then treated with nigericin. * *p* < 0.05; ** *p* < 0.01. All results are presented as the mean ± SD of the three independent experiments and were analyzed using Student’s *t*-test.

**Figure 7 ijms-21-05104-f007:**
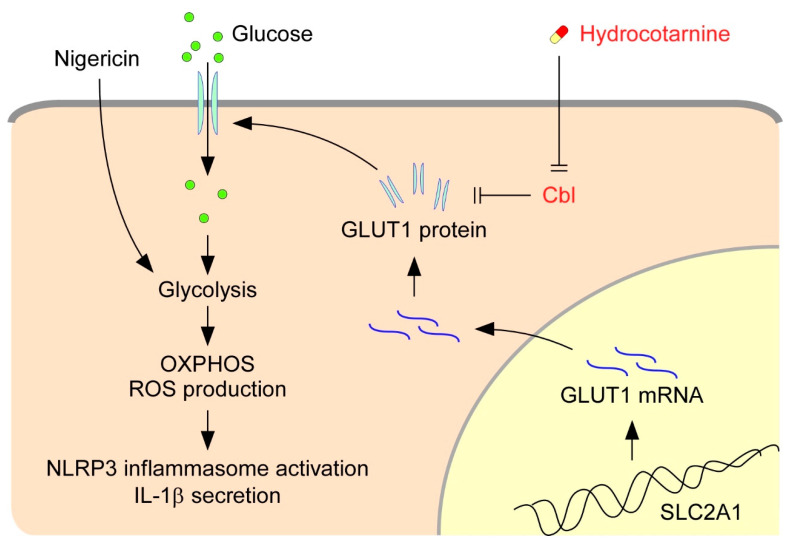
Model for the Cbl–GLUT1 axis-dependent negative regulation of NLRP3 inflammasomes. On the basis of our results, we proposed that Cbl reduces surface expression of GLUT1 protein and thereby contributes to suppression of the NLRP3 inflammasomes. GLUT1 protein encoded by the gene solute carrier family 2 member 1 (SLC2A1) is a major glucose transporter in macrophages that transports glucose from outside to inside cells. Intracellular glucose is an indispensable substrate for initiation of glycolysis. In response to stimulation of the NLRP3 agonist nigericin, glycolysis is activated and fuel oxidative phosphorylation (OXPHOS) activation occurs, which increases the generation of by-products from OXPHOS, known as cellular ROS, which then stimulate NLRP3 inflammasome activation and IL-1β secretion. GLUT1 expression is reduced by Cbl through post-transcriptional regulation. Reduction of GLUT1 by Cbl inhibits the cellular glucose uptake, which thus dampens the NLRP3 inflammasomes. Importantly, inhibition of Cbl with hydrocotarnine can remove the restraints on GLUT1 expression and enhance NLRP3 inflammasome activation.

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
