# Peer review of "Cbl Negatively Regulates NLRP3 Inflammasome Activation through GLUT1-Dependent Glycolysis Inhibition"

_ijms, 2020, doi:10.3390/ijms21145104_

Round 1
Reviewer 1 Report
In a previous report, the authors have demonstrated that the proto-oncogene Cbl can suppress NLRP3 inflammasome activation by inhibiting Pyk2-dependent ASC oligomerization. Here they report that in THP-1 cells, Cbl can also negatively regulate this inflammasome through GLUT1-dependent glycolysis inhibition. Indeed, cbl acts at the post-transcriptional level to decrease the amount of GLUT1, reducing therefore the cellular glucose uptake, and ensuing glycolysis-mediated mitochondrial ROS production.
My main concern about this manuscript is that the authors show that the NLRP3 inflammasome inducer nigericin promotes glycolysis, a required step, according to the authors, for inducing activation of this inflammasome. In BMDMs, it was described that the priming following TLR activation triggers a metabolic reprogramming but in THP-1 cells, the cells used in this study, do not require priming in the process of NLRP3 inflammasome activation. How do the author explain that nigericin activates glycolysis? Likewise, do other NLRP3 inflammasome inducers like ATP or MSU also stimulate glycolysis?
Other comments:
Line 126. The authors wrote: “we assessed glycolysis in Cbl-KO human embryo kidney (HEK) 293T cells which were shown previously to have higher levels of NLRP3 inflammasome activation than that of WT HEK293T cells [24]”. This statement is really surprising given that HEK293T cells does not express NLRP3 or ASC…
Figure 5. The amount of total GLUT1 should be also shown by WB.
Minor comments.
The authors should dampen their statement when they wrote that glycolysis and cellular ROS promotes NLRP3 inflammasome activation. I would say “enhance”.
Author Response
Reviewers' comments:
Reviewer 1
Comments and Suggestions for Authors
In a previous report, the authors have demonstrated that the proto-oncogene Cbl can suppress NLRP3 inflammasome activation by inhibiting Pyk2-dependent ASC oligomerization. Here they report that in THP-1 cells, Cbl can also negatively regulate this inflammasome through GLUT1-dependent glycolysis inhibition. Indeed, cbl acts at the post-transcriptional level to decrease the amount of GLUT1, reducing therefore the cellular glucose uptake, and ensuing glycolysis-mediated mitochondrial ROS production.
My main concern about this manuscript is that the authors show that the NLRP3 inflammasome inducer nigericin promotes glycolysis, a required step, according to the authors, for inducing activation of this inflammasome. In BMDMs, it was described that the priming following TLR activation triggers a metabolic reprogramming but in THP-1 cells, the cells used in this study, do not require priming in the process of NLRP3 inflammasome activation. How do the author explain that nigericin activates glycolysis? Likewise, do other NLRP3 inflammasome inducers like ATP or MSU also stimulate glycolysis?
Our response: We agree with the reviewer’s comments. In addition to nigericin, other NLRP3 inflammasome inducers including ATP and MSU can also stimulate glycolysis. Moon et al. showed that glycolysis assessed by the extracellular acidification rate (ECAR) was significantly increased by LPS and ATP stimulation compared with LPS alone in mouse peritoneal macrophages (Cell Rep. 2015 Jul 7;12(1):102-115). Finucane et al. showed that MSU increased ECAR and basal glycolysis in BMDMs (Sci Rep. 2019 Mar 11;9(1):4034). Therefore, glycolysis activation is likely to be induced by various NLRP3 agonists.
Although the mechanism of glycolysis activation by nigericin is still unknown, the mechanism of glycolysis activation by ATP was previously reported. Moon et al. showed that the expression levels of hexokinase 1 (HK1) were markedly increased in response to LPS and ATP, relative to LPS treatment alone in BMDMs. The induction of HK1 by ATP stimulation was regulated by mTORC1 activity (Cell Rep. 2015 Jul 7;12(1):102-115). HK is the first key enzyme in the glycolysis pathway and plays a vital role in the cellular uptake and utilization of glucose. Therefore, it is thought that ATP stimulation increases the expression level of HK1 through mTORC1-dependent mechanism, which consequently causes glycolysis activation. We speculate that nigericin may activate glycolysis through the same pathway as ATP; however, this hypothesis remains further investigation.
Other comments:
Line 126. The authors wrote: “we assessed glycolysis in Cbl-KO human embryo kidney (HEK) 293T cells which were shown previously to have higher levels of NLRP3 inflammasome activation than that of WT HEK293T cells [24]”. This statement is really surprising given that HEK293T cells does not express NLRP3 or ASC…
Our response: We agree with this comment. HEK293T cells do not express NLRP3 or ASC. Our previous studies to assess the levels of NLRP3 inflammasome activation were performed in NLRP3 inflammasome-reconstituted HEK293T cells (Cell Death Dis. 2018 Oct 31;9(11):1109). In the revised manuscript, we have modified the sentence to indicate that NLRP3 inflammasome components were reconstituted in HEK293T cells by transduction in the Results section as “Cbl-KO HEK293T cells with NLRP3 inflammasome reconstituted by transduction were shown previously to have higher levels of NLRP3 inflammasome activation than in NLRP3 inflammasome-reconstituted WT HEK293T cells [24]” (lines 126-128 in the revised version).
Figure 5. The amount of total GLUT1 should be also shown by WB.
Our response: We appreciate the reviewer’s comments. We added the new data, a WB analysis, to show the amounts of total GLUT1 in WT and Cbl-KO THP-1-derived macrophages or in hydrocotarnine-treated and untreated THP-1-derived macrophages in Fig. 5G and Fig. 5H in the revised manuscript, respectively. Consistent with the results of flow cytometry analysis in the original manuscript (Fig. 5E and Fig. 5F), WB analysis confirmed the increase of total GLUT1 protein in Cbl-KO cells by 2.7-fold compared with WT cells (Fig. 5G) and the increase in hydrocotarnine-treated cells by 1.6-fold compared with the untreated control (Fig. 5H). In the revised manuscript, we added the following statement in the Results section: “Similarly, the total GLUT1 protein in Cbl-KO cells increased 2.7-fold compared with WT cells (Figure 5G) or 1.6-fold in hydrocotarnine-treated cells compared with the untreated control (Figure 5H) as confirmed by immunoblot analysis” (lines 262-264 in the revised version).
Figure 5. Cbl reduces glucose transporter 1 (GLUT1) protein expression through post-transcriptional regulation. Flow cytometry plots of three experiments on revealing the surface expression of GLUT1 protein in (A) WT and Cbl-KO THP-1-derived macrophages or in (B) hydrocotarnine-treated and untreated THP-1-derived macrophages. Changes in GLUT1 mRNA expression were assessed in (C) WT and Cbl-KO THP-1-derived macrophages or in (D) hydrocotarnine-treated and untreated THP-1-derived macrophages. Flow cytometry plots of three experiments on revealing the expression of total GLUT1 protein in (E) WT and Cbl-KO THP-1-derived macrophages or in (F) hydrocotarnine-treated and untreated THP-1-derived macrophages. Immunoblot analysis of GLUT1 in (G) WT and Cbl-KO THP-1-derived macrophages or in (H) hydrocotarnine-treated and untreated THP-1-derived macrophages. *P < 0.05; **P < 0.01. All results are presented as the mean ± SD of three independent experiments and were analyzed using the Student’s t test.
Minor comments.
The authors should dampen their statement when they wrote that glycolysis and cellular ROS promotes NLRP3 inflammasome activation. I would say “enhance”.
Our response: We appreciate the reviewer’s comments. We have corrected the mistake. The expression “promote” was replaced by “enhance” (lines 163, 202, and 217 in the revised version).
Detailed modifications in the text:
Results:
Modification: “To examine whether an increase in glycolytic activity due to Cbl deficiency was a general effect, we assessed glycolysis in Cbl-KO human embryo kidney (HEK) 293T cells which were shown previously to have higher levels of NLRP3 inflammasome activation than that of WT HEK293T cells [24].” (lines 124-127 in the original version) was changed to “To examine whether an increase in glycolytic activity due to Cbl deficiency was a general effect, we assessed glycolysis in Cbl-KO human embryo kidney (HEK) 293T cells. Cbl-KO HEK293T cells with NLRP3 inflammasome reconstituted by transduction were shown previously to have higher levels of NLRP3 inflammasome activation than in NLRP3 inflammasome-reconstituted WT HEK293T cells [24].” (lines 126-128 in the revised version).
Modification: “promotes” (line 162 in the original version) was changed to “enhances” (line 163 in the revised version).
Modification: “promotes” (line 199 in the original version) was changed to “enhances” (line 202 in the revised version).
Addition: “Similarly, the total GLUT1 protein in Cbl-KO cells increased 2.7-fold compared with WT cells (Figure 5G), or 1.6-fold in hydrocotarnine-treated cells compared with the untreated control (Figure 5H), as confirmed by immunoblot analysis.” (lines 262-264 in the revised version).
Figures legends:
In Figure 3:
Modification: “promoted” (line 213 in the original version) was changed to “enhanced” (line 217 in the revised version).
In Figure 5:
Addition: “Immunoblot analysis of GLUT1 in (G) WT and Cbl-KO THP-1-derived macrophages or in (H) hydrocotarnine-treated and untreated THP-1-derived macrophages.” (lines 275-277 in the revised version).

Reviewer 2 Report
Lin et al investigate how Cbl, a ubiquitin ligase negatively regulates NLRP3. They have previously shown that Cbl inhibited Pyk2 kinase and thus oligomerization of ASC and that Cbl might be involved in regulation of glycolysis. In this study using KO cell lines and specific inhibitor they show how Cbl regulates metabolic pathways. They suggest that Cbl decreases surface expression of GLUT1 resulting in decreased glucose uptake and glycolysis leading to inhibition of NLRP3 inflammasome activation.
I find the study interesting. I wonder have the authors considered investigating whether Cbl regulates other inflammasomes via controlling GLUT1 expression. As 2-DG and NAC have been previously show to regulate the priming step of NLRP3 activation, lysate expression levels of NLRP3, pro-caspase-1 and pro-il-1beta should be included in Fig. 2C and 3C.
Please correct the words to:
Line 55 …. Molecular
93 …phosphorylate ASC
97 … mice exhibit
127 ….please indicate that NLRP3 inflammasome components were reconstituted in HEK293T cells by transduction as it is known that HEK293T by themselves do not express those components
Author Response
Reviewers' comments:
Reviewer 2
Comments and Suggestions for Authors
Lin et al investigate how Cbl, a ubiquitin ligase negatively regulates NLRP3. They have previously shown that Cbl inhibited Pyk2 kinase and thus oligomerization of ASC and that Cbl might be involved in regulation of glycolysis. In this study using KO cell lines and specific inhibitor they show how Cbl regulates metabolic pathways. They suggest that Cbl decreases surface expression of GLUT1 resulting in decreased glucose uptake and glycolysis leading to inhibition of NLRP3 inflammasome activation.
I find the study interesting. I wonder have the authors considered investigating whether Cbl regulates other inflammasomes via controlling GLUT1 expression. As 2-DG and NAC have been previously show to regulate the priming step of NLRP3 activation, lysate expression levels of NLRP3, pro-caspase-1 and pro-il-1beta should be included in Fig. 2C and 3C.
Our response: We appreciate the reviewer’s comments. Our previous studies examined whether Cbl regulates the AIM2 inflammasome. Cbl knockdown in THP-1 cells did not change the amount of poly(dAdT)-induced mature IL-1β, cleaved caspase-1, or ASC oligomers (Cell Death Dis. 2018 Oct 31;9(11):1109, Supplementary Figure S1). These results demonstrated that Cbl specifically suppresses the NLRP3 inflammasome, but not the AIM2 inflammasome. In addition, Moon et al. showed that a potent glycolysis inhibitor (2-deoxyglucose (DG)) did not significantly affect the secretion of IL-1β and IL-18 in wild-type BMDMs subjected to LPS and poly(dA:dT) treatment, which activates the AIM2 inflammasome pathway (Cell Rep. 2015 Jul 7;12(1):102-115). These findings indicate that Cbl and glycolysis are not involved in AIM2 inflammasome regulation. Therefore, we speculated that Cbl is not likely to regulate the AIM2 inflammasome via its control of GLUT1 expression. As for the role of the Cbl-GLUT1 axis in the regulation of other inflammasomes: this is unknown and remains under further investigation.
In addition, we added new data to show the levels of NLRP3, pro-caspase-1, and pro-IL-1β levels in the cells pretreated with 2-DG and NAC in Fig. 2C and Fig. 3C in the revised manuscript, respectively. Treatment of 2-DG or NAC does not change the levels of NLRP3, pro-casp-1, and pro-IL-1β. In the revised manuscript, we have modified the sentence in the Result section as follows: “The levels of mature IL-1β p17 and active caspase-1 (as assessed by the level of caspase-1 p20) in Cbl-KO cells were higher than in WT cells but were reduced by pretreatment with 2DG or glucose deprivation, while NLRP3, pro-caspase-1 and pro-IL-1β were unchanged (Figure 2C)” (lines 179-182 in the revised version). And “Neutralization of cellular ROS with NAC (Figure 3B) completely inhibited the expression of active caspase-1 p20 and mature IL-1β p17 (Figure 3C) and the secretion of IL-1β (Figure 3D) in both Cbl- KO and WT cells, while NLRP3, pro-caspase-1 and pro-IL-1β levels were unchanged.” (lines 209-212 in the revised version).
Figure 2. Cbl deficiency promotes NLRP3 inflammasome activation via upregulation of glycolysis. (A) ECAR of WT and Cbl-KO THP-1-derived macrophages under sequential treatment (dotted vertical lines) with or without 2DG and treated with nigericin. The change in ECAR between before and after nigericin treatment is calculated in the right panel. (B) OCR of WT and Cbl-KO THP-1-derived macrophages under sequential treatment (dotted vertical lines) with or without 2DG and treated with nigericin (B). The change in OCR between before and after nigericin treatment is calculated in the right panel. (C) Immunoblot analysis of caspase-1 (p20) and IL-1β (p17) in culture supernatants (SN) and NLRP3 inflammasome molecules and GAPDH in cell lysates (CL) of WT and Cbl-KO THP-1-derived macrophages that were treated with 2DG or glucose deprivation and then stimulated with nigericin. (D) Enzyme-linked immunosorbent assay of IL-1β in the supernatants of WT and Cbl-KO THP-1-derived macrophage cultures pretreated with or without 2DG and then stimulated with nigericin. **P < 0.01. All results are presented as the mean ± SD of three independent experiments and were analyzed using the Student’s t test.
Figure 3. Reactive oxygen species (ROS) production increased in Cbl-deficient THP-1-derived macrophages and promoted NLRP3 inflammasome activation. (A) The production of cellular ROS in WT and Cbl-KO THP-1-derived macrophages treated with or without nigericin for 30 min was measured using 2′,7′-dichlorodihydrofluorescein diacetate (H2-DCFDA). (B) The production of cellular ROS in WT and Cbl-KO THP-1-derived macrophages pretreated with or without NAC and stimulated with nigericin for 30 min was measured using H2-DCFDA. (C) Immunoblot analysis of caspase-1 (p20) and IL-1β (p17) in culture supernatants (SN) and NLRP3 inflammasome molecules and GAPDH in cell lysates (CL) of WT and Cbl-KO THP-1-derived macrophages that were pretreated with or without NAC and then stimulated with nigericin. (D) Enzyme-linked immunosorbent assay of IL-1β in the supernatants of WT and Cbl-KO THP-1-derived macrophage cultures pretreated with or without NAC and then stimulated with nigericin. **P < 0.01. All results are presented as the mean ± SD of three independent experiments and were analyzed using the Student’s t test.
Please correct the words to:
Line 55 …. Molecular
Our response: We have corrected the mistake. The expression “molecule” was modified as “molecular” (line 55 in the revised version).
93 …phosphorylate ASC
Our response: We have corrected the mistake. The expression “phosphorylating ASC” was modified as “phosphorylate ASC” (line 93 in the revised version).
97 … mice exhibit
Our response: We have corrected the mistake. The expression “mice exhibits” was modified as “mice exhibit” (line 97 in the revised version).
127 ….please indicate that NLRP3 inflammasome components were reconstituted in HEK293T cells by transduction as it is known that HEK293T by themselves do not express those components
Our response: We appreciate the reviewer’s comments. We have modified the sentence to indicate that NLRP3 inflammasome components were reconstituted in HEK293T cells by transduction in the Result section: “Cbl-KO HEK293T cells with NLRP3 inflammasome reconstituted by transduction were shown previously to have higher levels of NLRP3 inflammasome activation than in NLRP3 inflammasome-reconstituted WT HEK293T cells [24]” (lines 126-128 in the revised version).
Detailed modifications in the text:
Introduction:
Modification: “molecule” (line 55 in the original version) was changed to “molecular” (line 55 in the revised version).
Modification: “phosphorylating ASC” (line 93 in the original version) was changed to “phosphorylate ASC” (line 93 in the revised version).
Modification: “mice exhibits” (line 97 in the original version) was changed to “mice exhibit” (line 97 in the revised version).
Results:
Modification: “To examine whether an increase in glycolytic activity due to Cbl deficiency was a general effect, we assessed glycolysis in Cbl-KO human embryo kidney (HEK) 293T cells which were shown previously to have higher levels of NLRP3 inflammasome activation than that of WT HEK293T cells [24].” (lines 124-127 in the original version) was changed to “To examine whether an increase in glycolytic activity due to Cbl deficiency was a general effect, we assessed glycolysis in Cbl-KO human embryo kidney (HEK) 293T cells. Cbl-KO HEK293T cells with NLRP3 inflammasome reconstituted by transduction were shown previously to have higher levels of NLRP3 inflammasome activation than in NLRP3 inflammasome-reconstituted WT HEK293T cells [24].” (lines 126-128 in the revised version).
Modification: “Removal of cellular ROS with NAC (Figure 3B) completely inhibited the expression of active caspase-1 p20 and mature IL-1β p17 (Figure 3C) and the secretion of IL-1β (Figure 3D) in both Cbl- KO and WT cells.” (lines 206-208 in the original version) was changed to “Neutralization of cellular ROS with NAC (Figure 3B) completely inhibited the expression of active caspase-1 p20 and mature IL-1β p17 (Figure 3C) and the secretion of IL-1β (Figure 3D) in both Cbl- KO and WT cells, while NLRP3, pro-caspase-1 and pro-IL-1β levels were unchanged.” (lines 209-212 in the revised version).
Figures legends:
In Figure 2:
Modification: “(C) Immunoblot analysis of caspase-1 (p20) and IL-1β (p17) in culture supernatants of WT and Cbl-KO THP-1-derived macrophages that were treated with 2DG or glucose deprivation and then stimulated with nigericin.” (lines 193-195 in the original version) was changed to “(C) Immunoblot analysis of caspase-1 (p20) and IL-1β (p17) in culture supernatants (SN) and NLRP3 inflammasome molecules and GAPDH in cell lysates (CL) of WT and Cbl-KO THP-1-derived macrophages that were treated with 2DG or glucose deprivation and then stimulated with nigericin.” (lines 195-198 in the revised version).
In Figure 3:
Modification: “(C) Immunoblot analysis of caspase-1 (p20) and IL-1β (p17) in culture supernatants of WT and Cbl-KO THP-1-derived macrophages that were pretreated with or without NAC and then stimulated with nigericin.” (lines 217-219 in the original version) was changed to “(C) Immunoblot analysis of caspase-1 (p20) and IL-1β (p17) in culture supernatants (SN) and NLRP3 inflammasome molecules and GAPDH in cell lysates (CL) of WT and Cbl-KO THP-1-derived macrophages that were pretreated with or without NAC and then stimulated with nigericin.” (lines 220-223 in the revised version).

Round 2
Reviewer 1 Report
The referee's comments have been addressed in a satisfactory manner. I therefore recommend acceptance for publication.